# Starvation-resistant cavefish reveal conserved mechanisms of starvation-induced hepatic lipotoxicity

Macarena Pozo-Morales[1,*] , Ansa E Cobham[2,*] , Cielo Centola[2] , Mary Cathleen McKinney[2] , Peiduo Liu[3] , Camille Perazzolo[1], Anne Lefort[1], Frédérick Libert[1], Hua Bai[3] , Nicolas Rohner[2,4] , Sumeet Pal Singh[1]

**Starvation causes the accumulation of lipid droplets in the liver, a somewhat counterintuitive phenomenon that is nevertheless conserved from flies to humans. Much like fatty liver resulting from overfeeding, hepatic lipid accumulation (steatosis) during undernourishment can lead to lipotoxicity and atrophy of the liver. Here, we found that although surface populations of *Astyanax mexicanus* undergo this evolutionarily conserved response to starvation, the starvation-resistant cavefish larvae of the same species do not display an accumulation of lipid droplets upon starvation. Moreover, cavefish are resistant to liver atrophy during starvation, providing a unique system to explore strategies for liver protection. Using comparative transcriptomics between zebrafish, surface fish, and cavefish, we identified the fatty acid transporter slc27a2a/fatp2 to be correlated with the development of fatty liver. Pharmacological inhibition of slc27a2a in zebrafish rescues steatosis and atrophy of the liver upon starvation. Furthermore, down-regulation of FATP2 in Drosophila larvae inhibits the development of starvation-induced steatosis, suggesting the evolutionarily conserved importance of the gene in regulating fatty liver upon nutrition deprivation. Overall, our study identifies a conserved, druggable target to protect the liver from atrophy during starvation.**

## Introduction

Starvation is a severe form of malnutrition that occurs when an individual's intake of food is inadequate to meet their body's energy requirements. Prolonged starvation can cause permanent organ damage, stunted growth in children, and eventually death if left untreated (Hizli et al, 2007; Restellini et al, 2013). It is estimated that ~10% of the global population suffered from chronic undernourishment in 2021 (Roser & Ritchie, 2019; FAO et al, 2022) and that ~45% of deaths among children under the age of 5 yr are linked to undernutrition (Black et al, 2003; Bryce et al, 2005). This figure has been steadily increasing in recent years, partly because of factors such as conflicts and climate change (von Grebmer et al, 2022). Developing interventions aimed at improving starvation resistance is critically needed to fight against nutritional deficiency. Surprisingly, despite the widespread prevalence of starvation, there has been considerably more research focusing on preventing tissue damage resulting from overconsumption than from prolonged hunger.

Notably, the liver's function and health are compromised during starvation. Almost 90 yr ago, the pediatrician Dr. Cicely Williams recognized that severe protein deficiency in children leads to fatty liver, and named the condition "kwashiorkor" (Williams, 1935, 1983; Frenk et al, 1958; Ayonrinde, 2021). Subsequently, starvation has been shown to induce the accumulation of liver fat (steatosis) in Drosophila (Gutierrez et al, 2007), zebrafish (Xu et al, 2021; Pozo-Morales et al, 2023), and mammals, including mice (B'chir et al, 2018), minks (Bjornvad et al, 2004), cats (Center et al, 1993, 77), and humans (Sakada et al, 2006; Faragalla et al, 2022). Notably, cats suffering from sudden anorexia, because of decreased food availability or being secondary to a disease, develop feline hepatic lipidosis, which is the most common hepatobiliary disease in cats. The prognosis from feline hepatic lipidosis can be positive but only with appropriate and fast nutritional treatment (Valtolina & Favier, 2017) as mortality can be close to 100% without aggressive nutrition therapy (Jacobs et al, 1989). Hepatic steatosis damages the liver via a multitude of pathways, including increased ER stress, oxidative stress, mitochondrial dysfunction, and inflammation (Geng et al, 2021; Obaseki et al, 2024). Though considerable attempts are being made to develop interventions aimed at protecting the liver from lipotoxicity in response to high-fat diet or obesity, pharmacological interventions to protect the liver during starvation are non-existent.

To uncover potential strategies for protecting the liver, we turned to adaptation strategies observed in nature. Across the animal kingdom, numerous species have developed adaptations to cope with starvation, offering potential perspectives into strategies to combat its detrimental effects (McCue, 2010; Olsen et al, 2021). To

---

[1]IRIBHM, Université Libre de Bruxelles (ULB), Brussels, Belgium   [2]Stowers Institute for Medical Research, Kansas City, MO, USA   [3]Department of Genetics, Development, and Cell Biology, Iowa State University, Ames, IA, USA   [4]Department of Cell Biology & Physiology, University of Kansas Medical Center, Kansas City, KS, USA

Correspondence: nro@stowers.org; sumeet.pal.singh@ulb.be
*Macarena Pozo-Morales and Ansa E Cobham contributed equally to this work

gain mechanistic insights, we used a genetically tractable and naturally occurring model of starvation resistance—the *Astyanax mexicanus* model system (Jeffery, 2020; Cobham & Rohner, 2024). *A. mexicanus* are teleost freshwater fish found in the Sierra de El Abra region of Mexico. A unique feature of *A. mexicanus* is the presence of the surface and cavefish morphs that provides an ideal comparative model system for the study of starvation response. The surface populations reside in the rivers and have constant access to nutrition. In contrast, the cave populations of this species have adapted to survive in the nutrient-poor subterranean environment, which provides limited nutrition in the dry season. Seasonal flooding during the rainy season brings nutrition inside the cave environment, leading to large fluctuations in food availability during the year (Wilson et al, 2021; Medley et al, 2022). Thus, although the surface morphs display relatively normal vertebrate physiology, the cave morphs have adapted to conditions of nutrition variation, including resistance to long periods of starvation. We took advantage of this unique system to study the response of the liver to starvation and identified the reduced expression of the fatty acid uptake gene *slc27a2a/fatp2* allowing cavefish to prevent liver damage under starvation. Targeting this pathway in both fish and flies mitigates starvation-induced hepatic steatosis and protects the liver from lipotoxicity in zebrafish. This demonstrates that the identified pathway is evolutionarily conserved for over 400 million years, highlighting its potential as a druggable target.

# Results

### Higher starvation resistance of cavefish larvae as compared to surface fish larvae

It has been previously shown that cavefish of *A. mexicanus* fare better under starvation compared with their surface cousins when grown under ad libitum feeding conditions as they are able to accumulate large fat reserves as adults (Aspiras et al, 2015). However, whether this ability is independent of feeding and present at earlier developmental stages has not been rigorously addressed. Cavefish, like most fishes, are endowed with nutrition from the mother in the form of yolk. Upon consumption of the yolk, the larvae depend on exogenous nutrients for survival. It is known that cavefish eggs possess ~15% larger yolk depots compared with surface fish eggs (Huppop & Wilkens, 1991); however, visible yolk regression in cavefish completes by 5.5 days post-fertilization (dpf) as compared to 4.5 dpf in surface fish (Hinaux et al, 2011), implying that under laboratory conditions, *A. mexicanus* larvae enter starvation latest by 6 dpf. To assess whether cavefish larvae are more resilient to starvation as compared to surface fish larvae after this time point, we withheld nutrition and tracked daily survival. Surface fish larvae survived robustly until 10 dpf, but afterward showed a steep reduction in their survival curves (Fig 1A). In contrast, larvae of two parallelly evolved morphs of cavefish, Pachón and Tinaja, demonstrated more variability in their early survival, in line with previous observations of higher early embryonic mortality in certain laboratory strains; however, once they survived day 6 the cavefish larvae were able to survive up to 7 d longer than surface fish without food (Fig 1A).

Starvation is known to induce the accumulation of liver fat and liver atrophy in vertebrates (Center et al, 1993; B'chir et al, 2018). To test whether cavefish evolved a different tissue response to starvation, we quantified lipid droplet accumulation in the liver using Nile Red staining. In line with observations from other species, we observed an accumulation of lipid droplets in the liver of surface fish upon starvation, peaking at 7 dpf (Fig 1B and D). However, strikingly, we did not detect any noticeable increase in lipid droplets in the liver of cavefish during starvation (Fig 1B and D). Furthermore, we noticed that surface fish display starvation-induced liver atrophy, with the liver size decreasing by 48% from 6 to 12 dpf ($P$ = 0.002) (Fig 1C). However, the cavefish liver seemed to be more resilient to starvation-induced liver atrophy, with Pachón and Tinaja showing only a more modest reduction of 23% ($P$ = 0.15) and 28% ($P$ = 0.17), respectively, over the 6 d of starvation. To our knowledge, this is the first report of a model organism that is able to avoid starvation-induced hepatic steatosis.

### Liver atrophy upon accumulation of hepatic lipid droplets

We investigated the relationship between the accumulation of hepatic lipid droplets and liver atrophy upon starvation. It is known from non-alcoholic fatty liver disease that liver damage occurs when hepatic steatosis proceeds to steatohepatitis, characterized by the presence of inflammation (Younossi et al, 2023). However, to our knowledge, it is not known whether liver damage upon starvation progresses through a similar sequence of events. To investigate this, we turned to the zebrafish as a model system, where we and others have shown robust development of starvation-induced hepatic steatosis (Xu et al, 2021; Pozo-Morales et al, 2023), similar to surface fish. We used zebrafish with a WT genotype for the *slc16a6a* gene (Fig S1), as mutation in the gene has been shown to enhance lipid accumulation in the liver (Hugo et al, 2012). We visualized hepatic lipid droplets in the *Tg(fabp10a:EGFP)* reporter line, which marks hepatocytes with green fluorescence, using Nile Red staining of neutral lipids. We found that at 4 dpf, zebrafish display very little hepatic lipid droplets, whereas at 6 dpf, without exogenous feeding, zebrafish enter a fasting state with robust hepatic steatosis (Fig 2A) (Pozo-Morales et al, 2023). After 6 dpf, hepatic lipid droplets decreased (Fig 2A and C) and liver atrophied (Fig 2A and G). To quantify the macrophage presence as an indicator of inflammation, we took advantage of the double transgenic zebrafish line *Tg(fabp10a:mCherry)*; *Tg(mpeg1.1:EGFP)* to visualize hepatocytes and liver macrophages in vivo. Notably, because of the pH stability of mCherry, this combination allowed us to not only visualize macrophages, but also detect the presence of hepatic phagocytic debris in macrophages (Fig 2B). We found a significant increase in the number of macrophages in the liver at 7 and 8 dpf (Fig 2B and C), suggesting a transition from steatosis to steatohepatitis. Furthermore, the livers of 8 dpf fed animals, in which we have previously shown resolution of lipid droplets (Pozo-Morales et al, 2023), have baseline levels of the macrophage presence (Fig S2A and B; compare Fig S2B with Fig 2C), suggesting that the macrophage infiltration is specific to fasting.

Next, to investigate the relationship between hepatic lipid use and inflammation, we manipulated turnover of lipid droplets in the liver. We had previously shown that buffering of calcium signaling in

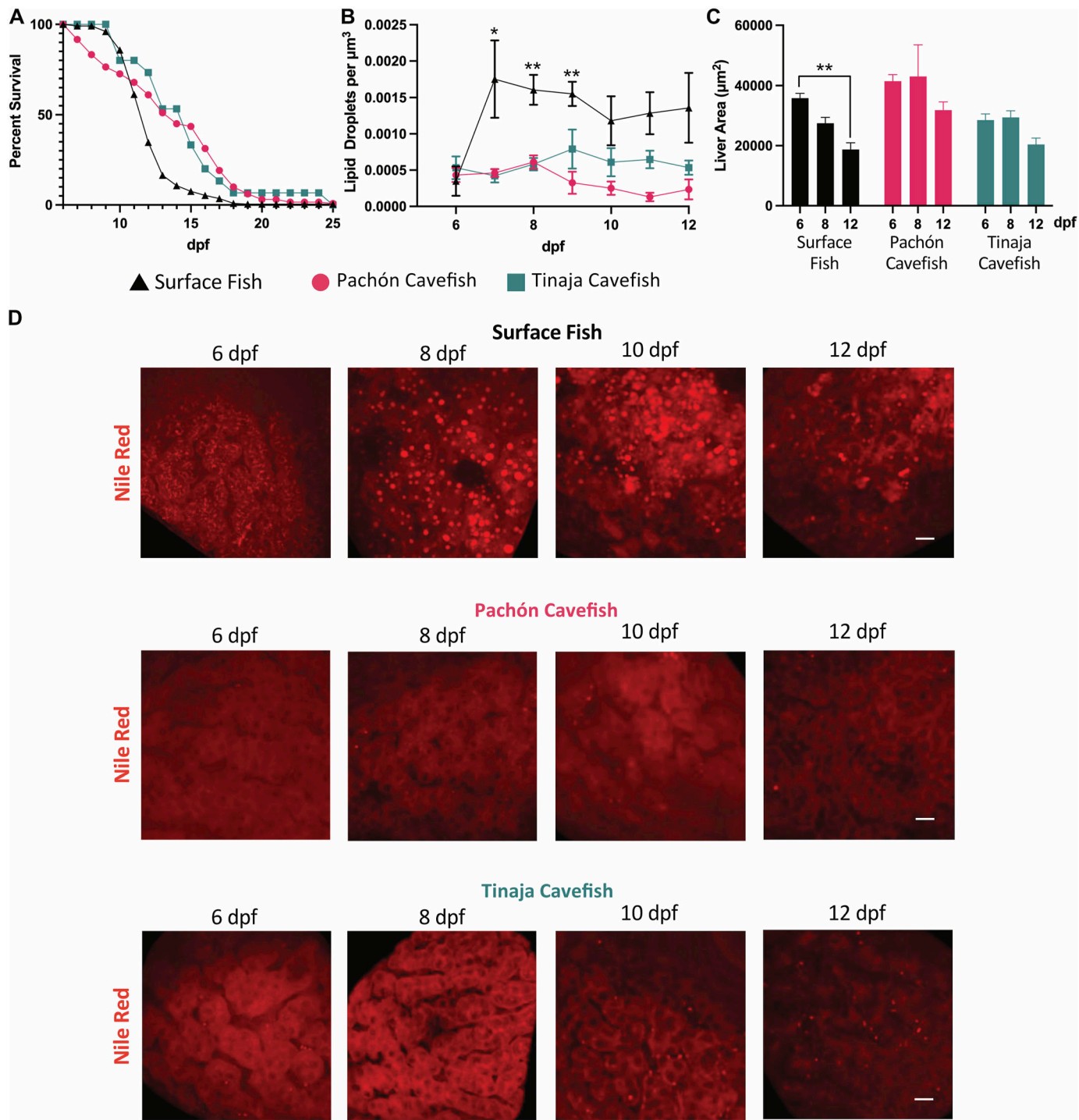

**Figure 1. Response of cavefish and surface fish larvae to starvation.**
**(A)** Survival curves for surface fish, Pachón cavefish, and Tinaja cavefish larvae upon starvation are shown by log-rank Kaplan–Meier plots with a 95% confidence interval. **(B)** Number of hepatic lipid droplets per unit volume from 6 to 12 dpf under fasting. **(C)** Mean ± SEM of liver size in surface fish, Pachón cavefish, and Tinaja cavefish at 6, 8, and 12 dpf without exogenous feeding. **P < 0.01. **(D)** Maximum-intensity projections of representative livers from surface fish, Pachón cavefish, and Tinaja cavefish stained with Nile Red (red) at 6, 8, 10, and 12 dpf without exogenous feeding. Scale bar = 10 µm.

the hepatocytes, accomplished by the hepatocyte-specific expression of a genetically encoded buffer of cytosolic calcium oscillations called *SpiCee* (Ros et al, 2020), initiates hepatic steatosis

at 4 dpf, in the fed state (Pozo-Morales et al, 2023). Here, we investigated the hepatic lipid content in the *Tg(fabp10a:SpiCee-mCherry)* animals upon fasting. In 6 dpf animals devoid of

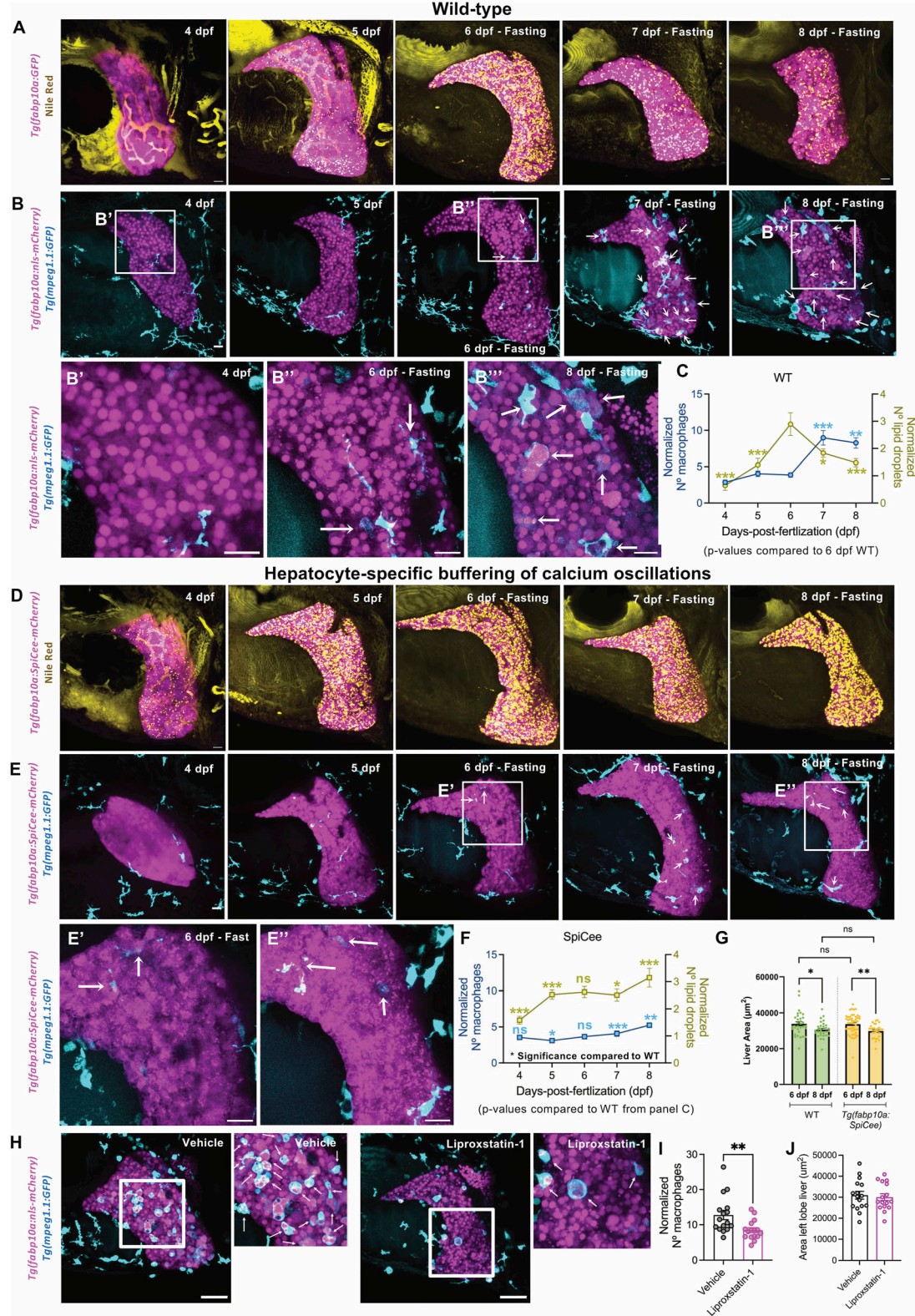

**Figure 2. Reduction in the size of liver upon accumulation of lipid droplets during starvation.**
**(A)** Maximum-intensity projections (MIP) of *Tg(fabp10a:GFP)* (referred to as WT) stained with Nile Red. Hepatocytes are false-colored in pink and lipid droplets in yellow. A timeline from 4 to 8 dpf of zebrafish larvae without exogenous food is presented. **(B)** Representative MIP images of timeline from *Tg(fabp10a:nls-mCherry); Tg(mpeg1.1: EGFP)* zebrafish from 4 to 8 dpf under the fasting condition. Arrows label macrophages with hepatocyte debris. A zoom of boxed regions is presented below the images. **(C)** Line trace representing lipid droplets and macrophages normalized to the liver area from 4 to 8 dpf. Comparisons with 6 dpf are represented. *P*-values: * < 0.05; ** <

exogenous food, hepatocytes from SpiCee-expressing and control animals had similar levels of lipid droplets (compare Fig 2B and C with Fig 2D and F). However, although the control animals reduce the droplets, SpiCee-expressing hepatocytes do not show a reduction in lipids until 8 dpf, maintaining a steady level of lipid droplets (Fig 2D and F). Strikingly, livers from SpiCee-expressing animals did not show a significant increase in liver macrophages from 6 to 8 dpf (Fig 2E and F). This suggests that reducing the turnover of lipid droplets protects the liver from inflammation. However, the liver in SpiCee-expressing animals did undergo atrophy at the same rate as the liver in WT animals (Fig 2G), suggesting that inhibiting lipid use cannot protect the liver from atrophy.

In accordance with the model where the lipid droplet turnover correlates with inflammation, several studies have suggested the role of lipid droplets as a store of potential pro-inflammatory molecules (Bailey et al, 2015; Feng & Stockwell, 2018; Jarc & Petan, 2020). Degradation of lipid droplets releases polyunsaturated fatty acids that can react with reactive oxygen species to generate cytotoxic lipid peroxidation molecules (Danielli et al, 2023). To evaluate whether lipid peroxides generated by lipid droplet degradation modulate the inflammatory response in the zebrafish liver, we treated *Tg(fabp10a:nls-mCherry);Tg(mpeg1.1:GFP)* animals with liproxstatin-1 (lipid peroxidation inhibitor) from 6 to 8 dpf under starvation. Reduction in lipid peroxidation decreased macrophages in the liver (Fig 2H and I), suggesting a role of lipids in steatohepatitis. However, liproxstatin-1 treatment could not protect the liver from atrophy, as the size of the liver between control and liproxstatin-1–treated animals was the same at 8 dpf (Fig 2J).

Overall, our data provide evidence that reduction in lipid flux or inflammation does not protect the liver from atrophy during starvation in the zebrafish larval stage.

### Inhibition of hepatic lipid accumulation protects the liver from atrophy during starvation

As the liver size reduces after the accumulation of lipid droplets, irrespective of the lipid flux or inflammatory status, the best strategy to protect the liver is potentially to avoid lipid droplet formation during starvation, as done by cavefish. With this, potentially cytotoxic lipid products are not accumulated in the hepatocytes. To understand how cavefish prevent the formation of lipid droplets upon starvation, we compared transcriptomes of livers from the fed and fasting state. We profiled the liver from 4 (fed) and 6 dpf (fasting) zebrafish larvae, and 5 (fed) and 7 dpf (fasting) surface fish and Tinaja cavefish larvae. Upon performing 2-D dimensionality reduction in the transcriptomes using multidimensional

scaling, we observed that the species was separated on the x-axis, suggesting species-specific responses to starvation (Fig 3A). However, the y-axis represented the fed-to-fast transition for the two species, suggesting a shared response to nutrition deprivation (Fig 3A). Notably, this underscores the fact the cavefish larvae also undergo starvation after 6 dpf.

A differential gene expression (DGE) analysis identified 367 genes to be modulated in all three groups (Fig 3B). Among these, 79 genes were up-regulated in the starvation state (Fig 3C). Gene ontology (GO) analysis revealed that these 79 genes were enriched in "regulation of translational initiation," "insulin-like growth factor signaling," "acylglycerol biosynthetic pathway," and "protein targeting to peroxisomes" (Fig 3D). We next focused on genes up-regulated upon starvation in zebrafish and surface fish, but not in cavefish. These 285 genes were enriched in "mitochondrion disassembly," "autophagosome assembly," "triglyceride catabolic process," "regulation of reactive oxygen species," and "lipid transport" (Fig 3E). Among the lipid transporters, we observed that the long-chain fatty acid transporter, *slc27a2a*, was differentially regulated in zebrafish and surface fish, but not in cavefish (Fig 3F).

The lipid transporter *slc27a2a* is the fish homologue of SLC27A2, also known as FATP2 (fatty acid transport protein 2), and belongs to the SLC27 family of lipid transporters (Anderson & Stahl, 2013).

To evaluate the role of *slc27a2a* during starvation, we treated zebrafish larvae with lipofermata, a specific inhibitor for SLC27A2 (Perez et al, 2020), from 4 to 6 dpf, and checked the impact of the drug at 6 dpf, and for 2 days post-treatment (dpt). At 6 dpf (0 dpt), livers from lipofermata-treated starved animals showed a dramatic reduction in the number of lipid droplets accumulated within the hepatocytes compared with vehicle-treated animals (Fig 3G and H). At 8 dpf (2 dpt), we observed an accumulation of lipid droplets in the liver, suggesting the inhibition is reversible (Fig S3A and B). Surface fish larvae treated with lipofermata showed the same trend: reduction in starvation-induced hepatic droplets after 48 h of treatment, and their accumulation after removal of the treatment (Fig S3D).

Next, we quantified liver size under the same treatment regimen and observed a striking increase in the size of the liver (Fig 3I). Notably, the increase in liver size was present at 6 dpf (0 dpt) and persisted at least until 8 dpf (2 dpt) (Fig S3A and C).

To identify the genes regulated by the lipofermata treatment, we performed DGE analysis of livers collected from 6 dpf fasting zebrafish treated with vehicle or lipofermata (Fig S4A). The analysis identified an up-regulation of 843 genes and a down-regulation of 1,067 genes in the liver upon lipofermata treatment (Table S1). Notably, genes related to cell cycle, including cyclins (*ccna2*, *ccnb1*, *ccnb2*, *ccnb3*), Ki-67 (*mki67*), and polo-like kinase 1, were up-

---

0.01, and *** < 0.001, ANOVA test followed by the post hoc Tukey test. **(D)** MIP of Nile Red staining of *Tg(fabp10a:SpiCee-mCherry)*, with hepatocytes false-colored in pink and lipid droplets in yellow. Representative images from 4 to 8 dpf fasting are presented. **(E)** Livers from 4 to 8 dpf from fasting *Tg(fabp10a:SpiCee-mCherry); Tg(mpeg1.1: EGFP)* are presented with MIP. Macrophages with hepatocyte debris are marked with arrows. A zoom of boxed regions is presented below the images. **(F)** Line trace representing lipid droplets and macrophages normalized to the liver area from 4 to 8 dpf in *Tg(fabp10a:SpiCee-mCherry)* animals. Comparisons with WT are shown. *P*-values: ns, not significant, * < 0.05, ** < 0.01, and *** < 0.001, *t* test or Mann–Whitney *U* test depending on the normality of the data. **(G)** Mean ± SEM of liver size in WT and *Tg(fabp10a:SpiCee-mCherry)* at 6 and 8 dpf without exogenous feeding. *P*-values: ns, not significant, * < 0.05, and ** < 0.01, *t* test. **(H)** Snapshots of livers at 8 dpf from *Tg(fabp10a:nls-mCherry); Tg(mpeg1.1:GFP)* fasting animals treated with 0.17% DMSO (vehicle) or 50 µM liproxstatin-1. Hepatocytes are false-colored in pink, and macrophages are false-colored in cyan. White arrows indicate macrophages with hepatocyte phagocytosis. **(I)** Mean ± SEM of the number of macrophages normalized by liver area in vehicle- and liproxstatin-1–treated animals. Each point represents a single animal. **P < 0.01, Mann–Whitney *U* test. **(J)** Mean ± SEM of the liver area in control and liproxstatin-1–treated animals at 8 dpf. Scale bar for all panels = 20 µm.

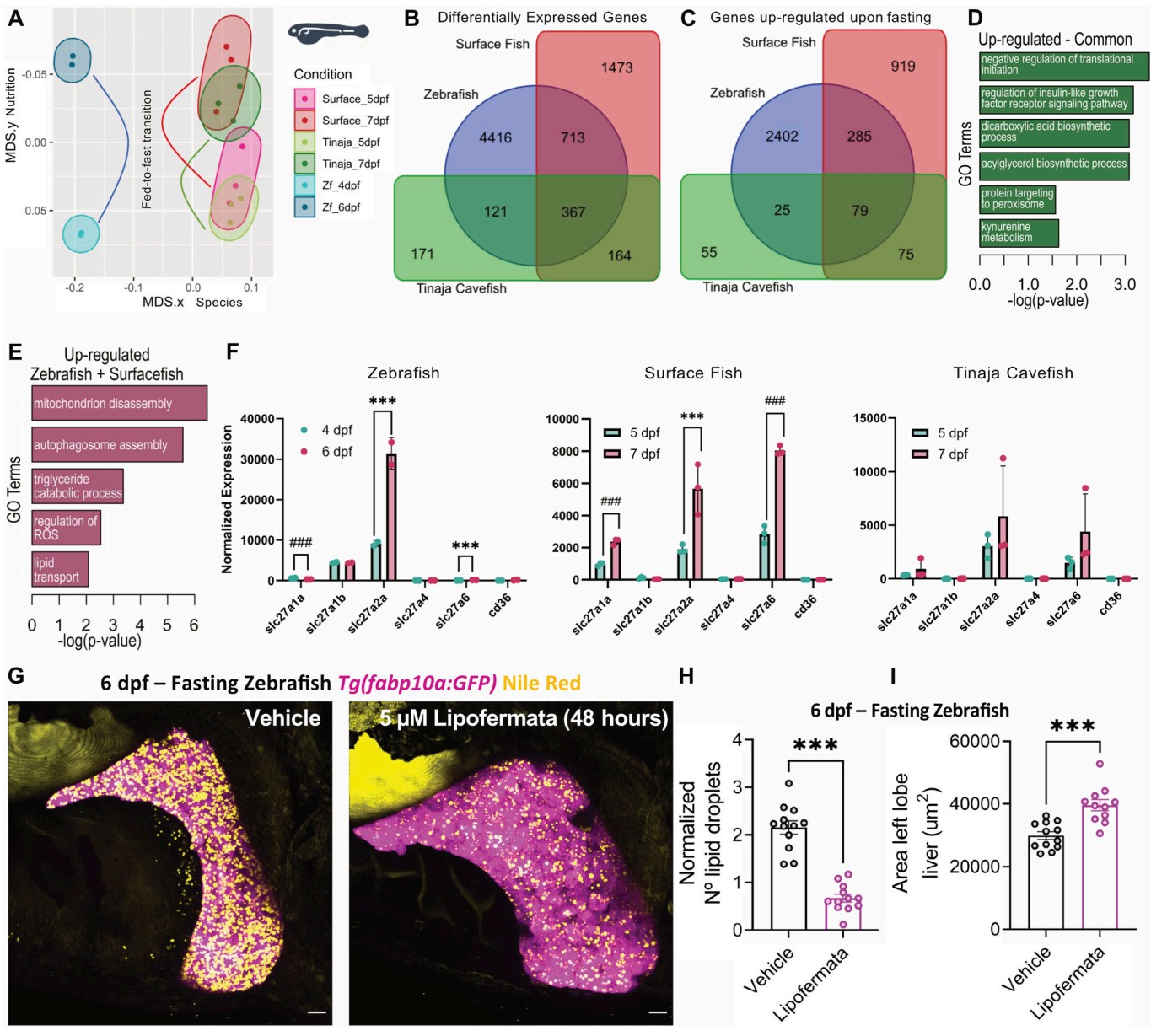

**Figure 3. *Slc27a2a* is responsible for starvation-induced lipid accumulation.**
**(A)** Multidimensional scaling plot of gene expression changes in the liver upon fasting in zebrafish, surface fish, and Tinaja cavefish. For zebrafish, livers from 4 dpf were compared with 6 dpf, whereas for surface and cavefish, the comparison was made between 5 and 7 dpf. **(B, C)** Venn diagram of differentially expressed genes (B) and genes up-regulated by fasting (C) for the three animals. **(D, E)** Gene ontology (GO) analysis for genes up-regulated by fasting in all the three animals (D) and for zebrafish and surface fish only (E). **(F)** Barplot displaying the changes in lipid transporters upon fasting. *** false discovery rate < 0.05 and log$_2$(fold change) > 1.5. ### (false discovery rate) < 0.05, but log$_2$(fold change) < 1.5. **(G)** Maximum-intensity projections of 6 dpf fasting *Tg(fabp10a:GFP)* (pink) with Nile Red staining (yellow) treated with 5 µM of lipofermata or 0.01% of DMSO (vehicle) from 4 to 6 dpf fast. Scale bar = 20 µm. **(H, I)** Barplot with the mean ± SEM of the number of lipid droplets per liver (H) and liver size (I) in vehicle- and lipofermata-treated animals. Each point represents a single animal. ns, not significant, ***$P < 0.001$, *t* test.

regulated upon lipofermata treatment (Fig S4B). The increase in cell cycle–related genes corroborates the increase in liver size in lipofermata-treated animals. Furthermore, lipofermata treatment led to a reduction in the expression of cytokines, including TNF superfamily member 10 (*tnfsf10*), chemokine ligands (*ccl25b*, *cxcl12a*, *cxcl18b*), and BMP/TGF-beta pathway ligands (*bmp2a*, *tgfb1a*, *tgfb1b*) (Fig S4C), suggesting the lowering of the inflammatory molecules in the milieu. However, the number of macrophages in the

liver did not change upon lipofermata treatment (Fig S4D). Overall, the transcriptional profiling of livers from lipofermata-treated animals suggests that the organ can initiate cell replication in the fasting state, thereby being capable of maintaining its size.

Treatment with lipofermata did not alter the body size of the animal (Fig S5A and B) or cause persistence of yolk (n = 12/12, Fig S5A). Furthermore, lipofermata-treated animals did not display an

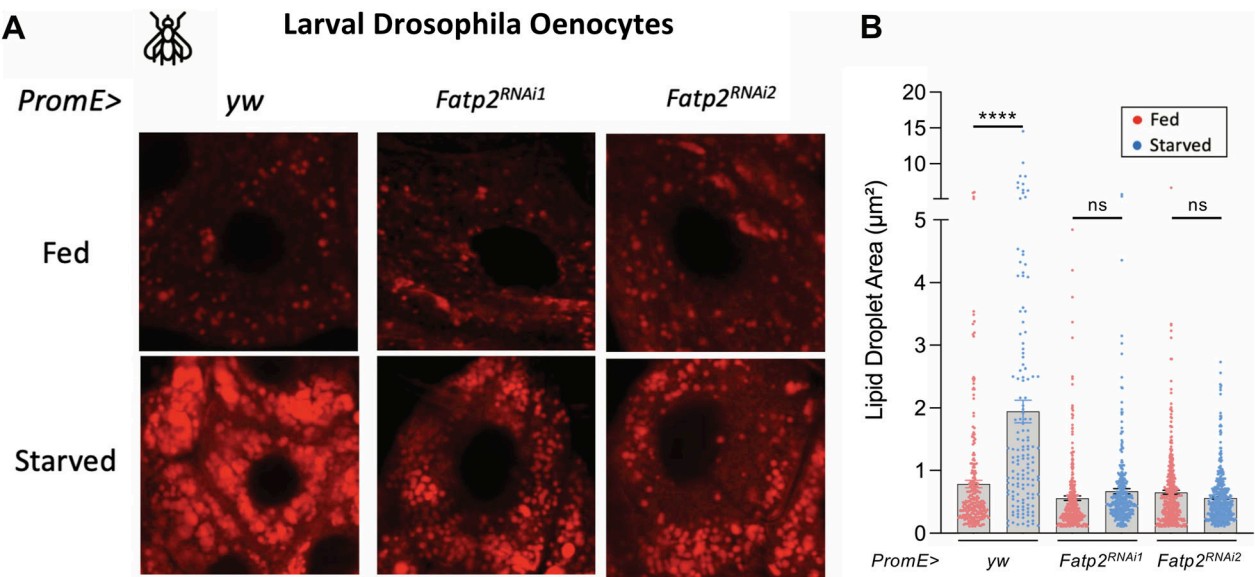

**Figure 4. FATP2 is an evolutionarily conserved regulator of starvation-induced lipidosis.**
**(A)** Confocal images of Drosophila oenocytes from fed and starved larvae. Control or oenocyte-specific FATP2 RNAi animals were evaluated. **(B)** Barplot comparing the lipid droplet area in the fed or fasting condition in control and FATP2 RNAi animals. ns > 0.05, ****P < 0.0001, ANOVA test followed by the post hoc Tukey test.

increase in the size of the liver upon feeding (Fig S5C and D), suggesting that the impact of lipofermata is particularly notable on the liver of the fasting animals.

Our experiments suggest that inhibition of fatty acid uptake can reduce accumulation of lipid droplets in the liver, which in turn can reduce liver atrophy upon starvation. However, the inhibition of fatty acid uptake upon starvation in zebrafish does not improve starvation resistance (Fig S5E), suggesting additional mechanisms of increased starvation resistance might play a role in cavefish.

### SLC27A2/FATP2 is an evolutionarily conserved regulator of starvation-induced hepatic steatosis

To test the evolutionary conservation of SLC27A2/FATP2 in starvation-induced steatosis, we turned to Drosophila, which shows the robust accumulation of lipid droplets in oenocytes, the evolutionary homologues of hepatocytes in the fly (Gutierrez et al, 2007). Using tissue-specific RNAi, we knocked down the Drosophila homologue of FATP2. Two independent siRNA lines were used. Upon starvation, the oenocytes from control animals displayed an accumulation of lipid droplets (Fig 4A and B). However, RNAi-mediated knockdown of FATP2 significantly reduced the number of lipid droplets (Fig 4A and B). This indicates that SLC27A2/FATP2 is required for the accumulation of lipid droplets in the Drosophila larval oenocytes upon starvation.

## Discussion

In this work, we report the importance of SLC27A2/FATP2, a fatty acid transporter, in initiating starvation-induced hepatic steatosis. The gene is up-regulated in the liver upon fasting in the larvae of zebrafish and *A. mexicanus* surface fish morph (Fig 3F). The increase

in gene expression allows uptake and accumulation of fatty acids in the hepatocytes, as pharmacological inhibition of the gene in zebrafish leads to a reduction in hepatic steatosis (Fig 3G and H). Moreover, knockdown of the gene in Drosophila oenocytes, the fly homologue of hepatocytes, prevents the collection of lipid droplets in the starving fly larvae (Fig 4). Thus, the role of FATP2 in the process is conserved from flies to fish, across ~400 million years of evolution (Wheeler & Brändli, 2009).

In contrast to Drosophila, zebrafish, and *A. mexicanus* surface fish morph, the cavefish morph of *A. mexicanus* does not display starvation-induced hepatic steatosis (Fig 1B and D). Correspondingly, cavefish larvae do not display an up-regulation in the expression of *slc27a2a* in the liver upon fasting (Fig 3F). Forgoing the accumulation of lipid droplets in the liver protects the cavefish liver from lipotoxicity and atrophy upon nutrition deprivation (Fig 1C). Mimicking this condition in zebrafish by pharmacological inhibition of *slc27a2a* rescues the loss of liver size upon fasting (Fig 3G). Notably, the antagonism of *slc27a2a* increases the expression of cell cycle–related genes in the liver, while reducing the expression of inflammatory cytokines (Fig S4B and C). Particularly, the treatment reduces the expression of TGF-ß ligands, which have been shown to induce hepatic fibrosis, a key step in irreversible liver damage (Dooley & ten Dijke, 2012). Thus, our study shows that the optimal solution to starvation-induced liver atrophy, identified by naturally starvation-resistant cavefish, is to prevent the accumulation of hepatic lipid droplets.

Our study, however, does not identify the molecular link between starvation and enhancement of *slc27a2a* expression. Endocrine signaling could be a potential regulator of initiating starvation response in the liver. Hepatocytes are sensitive to insulin and glucagon, which are released from the pancreas in response to feeding and fasting, respectively. Changes in the levels of circulating insulin and/or glucagon hormones could be sensed as a fed-

to-fast transition in the hepatocytes. Accordingly, work in Drosophila has identified a crosstalk between the fat body and oenocytes via the insulin signaling pathway as being responsible for the starvation-induced accumulation of lipid droplets (Chatterjee et al, 2014). Notably, cavefish carry a hypomorphic mutation in the insulin receptor (Riddle et al, 2018), which could reduce its ability to sense such a fed-to-fast transition. In the future, it would be of interest to investigate the link between insulin signaling and *slc27a2a* expression levels. Furthermore, the search for regulators of starvation-induced hepatic steatosis can be aided by performing quantitative trait locus (QTL) mapping in *A. mexicanus* (O'Quin et al, 2013; Stockdale et al, 2018). This is made possible by the fact that cavefish and surface fish are interfertile, which allows the generation of cave/surface genetic hybrids, providing a unique platform to identify potential regulators of the phenomenon.

In addition to *slc27a2a*, previous work in zebrafish has implicated CD36 as a regulator of starvation-induced hepatic steatosis (Xu et al, 2021). However, the expression of *cd36* in the liver of fasting or fed fish larvae is an order of magnitude lower than the expression level of *slc27a2a* (Fig 3F). Furthermore, the oenocyte-specific knockdown of FATP2/SLC27A2 in the fly larvae highlights its importance in a tissue-specific manner (Fig 4). Thus, we posit SLC27A2 as the major actor in fatty acid transport in hepatocytes during fasting.

Though inhibition of *slc27a2a* can protect the zebrafish liver from atrophy, it does not improve the organism's starvation resistance (Fig S5E). Metabolic plasticity in other organs might play a critical role in the enhanced starvation resistance of cavefish larvae (Fig 1A). It is known that cavefish larvae reduce their average oxygen consumption upon starvation (Bilandžija et al, 2020). In Drosophila, enhanced starvation resistance is related to changes in the basal metabolic rate (Brown et al, 2019). Future investigations in the cavefish's ability to survive long periods of famine are required; however, our work does highlight the utility of comparative evolutionary analysis in uncovering regulators of organ atrophy upon starvation.

# Materials and Methods

### *A. mexicanus* husbandry

Surface, Tinaja, and Pachón morphs of *A. mexicanus* were raised in polycarbonate recirculating aquaculture racks (Pentair Aquatic Eco-Systems), with a 14:10-h light:dark photoperiod. Each rack system is equipped with mechanical, chemical, and biological filtration and UV disinfection. Water quality parameters are monitored daily as described in previous studies (Xiong et al, 2022). Fish were housed at a density of ~2 fish per liter. Fish embryos spawned at the same time from different parent tanks were mixed to reduce effects from specific backgrounds. Only embryos screened as healthy were raised and randomly selected for the experiment. Embryos up to 5 d after fertilization were maintained at 24.5°C in embryonic E2 media. Animal husbandry followed protocol 2021-122 approved by the Institutional Animal Care and Use Committee of the Stowers Institute for Medical Research.

Housing conditions meet federal regulations and are accredited by AAALAC International.

### Survival curves for *A. mexicanus*

At 5 dpf, larvae were randomly divided into two groups. Each group included 10 biological replicates, and 20 embryos as one biological sample. One group, as the controls, were fed with *Artemia nauplii* (Brine Shrimp Direct) once per day. The other, as the starvation group, were not fed. The fish were identically treated in all other aspects, and the number of living fish was recorded daily. Dead fish were removed each day. The experiment was repeated three times independently.

### Nile Red staining, confocal imaging, and analysis for *A. mexicanus*

Nile Red staining was performed as previously described in Xiong et al (2022) with some modifications. Briefly, a stock solution of Nile Red (#N-1142; Invitrogen) was made by dissolving in acetone to a concentration of 500 $\mu$g/ml and stored in the dark at 4°C. A final working solution, diluted to 1/300 in embryonic E2 media, was prepared. For the staining procedure, 10 larvae per well were incubated in 12-well cell culture plates (#CLS3513; Corning) containing 2 ml in the working solution for 1 h at 25°C and protected from light. After incubation, stained larvae were euthanized with 500 mg/l MS-222, washed with PBS, and then fixed in 4% PFA for 20 min at room temperature before imaging. Samples were placed on a glass-bottomed FluoroDish, and images were captured using a Nikon Eclipse Ti2 inverted microscope equipped with Yokogawa Spinning Disk Confocal with a 40x/1.15 NA objective. Nile Red was excited with a 561-nm laser, and emission was collected between 579 and 631 nm. Z-stack images were acquired with 2048 × 2044 pixel size and step size of 1 $\mu$m.

Image quantification was performed in Imaris (Oxford Instruments). We first manually segmented the liver to find the volume and mask the lipid fluorescence channel. Lipid droplets were created using the spots function set to 0.8 $\mu$m seed size. The liver area was measured in ImageJ (Schindelin et al, 2012) by outlining the whole liver in a maximum projected image with the polygon selection tool.

### RNA sequencing of *A. mexicanus* livers

For RNA-Seq, RNA was isolated from livers at the corresponding stages. For this, livers from 40 larvae/replicate were dissected and collected in the lysis buffer from ReliaPrep RNA Miniprep Systems (Z6110; Promega). RNA was isolated from the lysed tissue following the manufacturer's instruction. cDNA synthesis, library preparation, and Illumina sequencing were performed according to the manufacturer's instructions using TruSeq Stranded mRNA Preparation Kit for Illumina (#20020594; Illumina). Purified libraries were quantified using a Qubit fluorometer (Life Technologies), and the quality was assessed on Agilent Bioanalyzer (Agilent Technologies). Sequencing was performed on the Illumina NextSeq 500 instrument as High Output with single-end 75-bp reads. After sequencing, raw reads were demultiplexed into a Fastq format using Illumina bclfastq2 v2.18.

## Mapping of transcriptome from *A. mexicanus* livers and comparison with zebrafish transcriptome

Raw reads were mapped using STAR aligner v2.7.3a (Dobin et al, 2013) against the UCSC genome astMex_2.0, using Ensembl 102 gene models for annotations. Transcript abundance was quantified using RSEM v1.3.0. For normalization, DGE analysis, and gene ontology (GO) analysis, iDEP version 0.951 (Ge et al, 2018) was used with default parameters. For DGE, a fold change of 2 and a false discovery rate of 0.1 were used for cutoff. Data for zebrafish liver transcriptomes from the fed and fasting conditions were taken from our previous publication (Pozo-Morales et al, 2023). Common and uniquely differentially expressed genes were compared using an online Venn Diagram tool (Draw Venn Diagram, 2023).

## Zebrafish strains and husbandry

Wild-type or transgenic zebrafish of the outbred AB strain were used in all experiments. Zebrafish were raised under standard conditions at 28°C. Animals were chosen at random for all experiments. Zebrafish husbandry and experiments with all transgenic lines will be performed under standard conditions in accordance with institutional (Université Libre de Bruxelles) and national ethical and animal welfare guidelines and regulations, which were approved by the Ethical Committee for Animal Welfare (CEBEA) from the Université Libre de Bruxelles (protocols 854N–865N). The following transgenic zebrafish strains were used in this study: *Tg(fabp10a:SpiCee-mCherry)*[ulb16] (Pozo-Morales et al, 2023), *Tg(mpeg1.1: EGFP)*[gl22] (Ellett et al, 2011), *Tg(fabp10a:nls-mCherry)*[mss4] (Mudbhary et al, 2014), and *Tg(fabp10a:EGFP)*[as3] (Her et al, 2003). The developmental stages of zebrafish used in experiments are prior to sex specification. All zebrafish were healthy and not involved in previous procedures.

## Genotyping of the *slc16a6a* gene

The genotyping of the *slc16a6a* gene was performed on cDNA prepared from the livers of 6 dpf *Tg(fabp10a:EGFP)* zebrafish. For this, livers from 30 larvae/replicate were dissected and collected in the lysis buffer from ReliaPrep RNA Miniprep Systems (Z6110; Promega), followed by isolation of total RNA from the lysate following the manufacturer's instruction. cDNA synthesis was carried out using ProtoScript II First Strand cDNA Synthesis Kit (E6300S; NEB) following the manufacturer's instruction. The synthesized cDNA was used to amplify the *slc16a6a* gene using the following primers: Forward: 5′-TGAACTCCTGGCAGTGAACA-3′ and Reverse: 5′-TTCCAGTACATTCAGTTTGAA-3′. The amplified product was gel-purified and cloned into CloneJET PCR Cloning Kit (K1231; Thermo Fisher Scientific). The cloned *slc16a6a* coding sequence was genotyped using Sanger sequencing.

## Pharmacological treatments

Stocks for lipofermata (HY-116788; MedChem) and Liproxstatin-1 (SML1414; Sigma-Aldrich) were prepared in DMSO and stored at −80°C. The working concentration for lipofermata was 5 $\mu$M (zebrafish) and 2 $\mu$M (surface fish), and for Liproxstatin-1 was 50 $\mu$M.

Incubation was performed in a petri dish sealed with parafilm to avoid evaporation. The petri dish was kept in the dark during the treatment. DMSO was used as a control. For treatment of fed animals with lipofermata, the animals were fed with rotifers starting at 5 dpf. On 6 and 7 dpf, the larvae were fed with rotifers during the day (8 h) and transferred to the E3 medium containing lipofermata or vehicle for overnight treatment (16 h). At 8 dpf, after two rounds of drug treatment, the animals were used for imaging.

## Survival curves for zebrafish larvae treated with lipofermata

At 4 dpf, larvae were randomly divided into two groups. Each group included 30 animals. One group was treated with 5 $\mu$M lipofermata, whereas the other group was treated with vehicle (0.01% DMSO). At 6 dpf, the larvae were dispensed individually into a 24-well plate with 1 ml E3 medium. Each subsequent day, half of the E3 medium (500 $\mu$l) was exchanged with fresh medium, and the number of living fish was recorded. Dead fish were removed each day. The experiment was repeated two times.

## RNA sequencing and analysis from zebrafish

For RNA-Seq, mRNA was isolated from livers of 6 dpf fasting zebrafish upon lipofermata treatment for 48 h. For this, we followed the protocol exactly as outlined in our previous work (Pozo-Morales et al, 2023). For DGE analysis and gene ontology analysis, iDEP version 0.96 (Ge et al, 2018) was used with default parameters. For DGE, a false discovery rate of 0.1 was used for cutoff.

## Imaging of the zebrafish liver macrophages

Animals were anesthetized in 0.02% tricaine methanesulfonate (MS-222; E10521; Sigma-Aldrich), mounted in 1% low-melt agarose containing 0.02% Tricaine MS-222 (50080; Lonza), and imaged on a glass-bottomed FluoroDish (FD3510-100; World Precision Instruments) using a Zeiss LSM 780 confocal microscope. Livers were imaged using a 40x/1.1N.A. water correction lens. Samples were excited with 488 nm for *Tg(mpeg1.1:EGFP)* and 543 nm for *Tg(fabp10a: nls-mCherry)* or *Tg(fabp10a:SpiCee-mCherry)*. The imaging frame was set at 1024 × 1024 pixels, and the distance between confocal planes was set at 3 $\mu$m for Z-stack to cover, on average, a thickness of 100 $\mu$m. Analysis was performed in Fiji (Schindelin et al, 2012). Macrophages in contact with hepatocytes were counted manually. The liver area was calculated by outlining the region using the magic wand area selection tool of a Z-stack projection, followed by the "Analyze" – "Measure" (Area) command. The normalized number of macrophages was calculated using the next formula:

$$\frac{Total\ Number\ of\ Macrophages}{Area\ of\ the\ liver} * 10^{-4}$$

## Nile Red staining, confocal and brightfield imaging, and analysis for zebrafish

Imaging of the lipid droplets in the zebrafish liver was performed exactly as mentioned in our previous publication (Pozo-Morales

et al, 2023), which includes the timeline of hepatic lipid droplets in the wild-type condition from 4 to 6 dpf. In brief, the Nile Red stock was prepared by dissolving 500 µg/ml of Nile Red (Art. No. 7726.2; Carl Roth) in acetone and filtering the solution with a 0.2-µm filter. The stock solution was stored at 4°C in the dark. On the day of imaging, a fresh working solution of 1/300 in the embryonic E2 medium was prepared. The larvae were incubated in the working solution for 1 h at 28°C, protected from light. After incubation, larvae were washed once with E2 medium. Animals were anesthetized with 0.02% MS-222, mounted in 1% low-melt agarose containing 0.02% of MS-222, and imaged on a glass-bottomed FluoroDish using a Zeiss LSM 7890 confocal microscope. Livers were imaged using a 40x/ 1.1N.A. water correction lens. The imaging frame was set at 1024 × 1024 pixels, and the distance between confocal planes was set at 2 µm for Z-stack to cover, on average, a thickness of ~100 µm. Samples were excited with 488 nm, and a spectral detector was used to collect fluorescence corresponding to EGFP (500–508 nm) and neutral lipids labeled by Nile Red (571–579 nm).

Analysis was performed in Fiji (Schindelin et al, 2012). Lipid droplets were outlined manually with a circular selection tool. Only lipid droplets (marked with red fluorescence) present within hepatocytes (marked with green fluorescence) were counted. The area of the hepatocyte region was calculated by outlining the region using the magic wand area selection tool, followed by the "Analyze" – "Measure" (Area) command.

For measurement of liver size in *Tg(fabp10:EGFP)* animals, the protocol outlined above was followed, except Nile Red staining was not performed. Thus, only the EGFP signal was collected by excitation with a 488-nm laser and collection using a band-pass filter of 493–598 nm. Using Fiji, the confocal stacks were projected onto a maximum-intensity projection image, and the liver was outlined using the magic wand area selection tool, followed by the "Analyze" – "Measure" (Area) command.

Brightfield images of zebrafish larvae were collected using a Leica S8 APO stereo microscope. For this, the larvae were anesthetized with 0.02% MS-222 and mounted in 1% low-melt agarose containing 0.02% of MS-222. Length was measured from the tip of the mouth to the end of the notochord (the caudal fin was not included in the length measurement).

### Drosophila stocks and husbandry

Flies were maintained at 25°C, 60% relative humidity, and 12-h light/dark cycles. Larvae were reared on the standard cornmeal- and yeast-based diet (0.8% cornmeal, 10% sugar, and 2.5% yeast). For starvation treatment, early L3 larvae (~72 h after egg laying) were kept with Kimwipe soaked with 1x phosphate-buffered saline for 16–18 h. Fly stocks used in this study were as follows: PromE-Gal4 (#65405; BDSC), Fatp2 RNAi (#55271; #1-BDSC, #44658; #2-BDSC). ywR flies (a gift from Eric Rulifson) were crossed to PromE-Gal4 as a control.

### Nile Red staining, confocal imaging, and analysis for Drosophila oenocytes

Larva was dissected in Schneider's Drosophila medium (#217-20024; Thermo Fisher Scientific) (normal fed larvae) or in 1x phosphate-

buffered saline (1x PBS) (starved larvae). The dissected larvae were fixed with 4% PFA (#15710; Electron Microscopy Sciences) for 20 min at room temperature. The stock Nile Red was prepared by dissolving 1 mg of Nile Red (#N1142; Invitrogen) in 1 ml of DMSO. The larval tissues were incubated in 2 µg/ml Nile Red working solution for 1 h at room temperature, protected from light. The samples were then washed and mounted on the slides using ProLong Diamond Antifade Mountant (#P36961; Thermo Fisher Scientific). The lipid droplets were imaged with FV3000 Confocal Laser Scanning Microscope (Olympus). The imaging frame was set at 1024 × 1024 pixels. Samples were excited with 555 nm, and the range of 580–610 nm was used to detect the Nile Red signal. The images were analyzed by Olympus cellSens Dimensions software. The lipid droplets were identified by using the region of interest and adjusting the fluorescent threshold to include the lipid droplets inside the oenocytes. The area is calculated automatically by Olympus cellSens Dimensions software after identifying the lipid droplets.

### Statistical analysis

Statistical analysis was performed using GraphPad Prism software (version 9.3.1; GraphPad Software). The test used for comparison is mentioned in the figure legends of the respective graphs. For all analyses, data were tested for outliers and normal distribution. For non-normal distributed data (or if the test for normal distribution could not be performed because of the small sample size), a non-parametric test was used. No data were excluded from the analysis. Blinding was not performed during the analysis.

## Data Availability

All data needed to evaluate the conclusions in the article are present in the article and/or the Supplementary Materials. Additional data related to this article may be requested from the authors. The sequencing data are available at NCBI Gene Expression Omnibus under GSE244648 (cavefish and surface fish liver during fasting) and GSE252997 (zebrafish liver upon lipofermata treatment).

## Supplementary Information

## Acknowledgements

We thank the members of IRIBHM Fish Facility and M Martens and JM Vanderwinden from the Light Microscopy Facility for technical assistance at ULB. We are grateful to the cavefish facility at the Stowers Institute for Medical Research for cavefish husbandry support. The authors would also like to thank Amanda Lawlor from the Stowers Sequencing and Discovery Genomics team for help regarding library preparation and sequencing of cavefish samples. We thank Bloomington Drosophila Stock Center (supported by NIH P40OD018537) for fly stocks. We thank Eric Rulifson for sharing the yw fly stock. Fonds de la Recherche Scientifique (FNRS) grants 40005588

and 40013427 (to SP Singh), Stowers Institute grants (to N Rohner), National Institute of Health (NIH) grants 1DP2AG071466-01 and R24OD030214 (to N Rohner), and R01AG058741 and R01AG075156 (to H Bai), and National Science Foundation (NSF) CAREER grant 2046984 (to H Bai).

## Author Contributions

M Pozo-Morales: investigation, visualization, and methodology.
AE Cobham: investigation, visualization, and methodology.
C Centola: investigation, visualization, and methodology.
MC McKinney: visualization.
P Liu: investigation, visualization, and methodology.
C Perazzolo: investigation.
A Lefort: investigation.
F Libert: investigation.
H Bai: supervision, visualization, methodology, and writing—review and editing.
N Rohner: conceptualization, supervision, funding acquisition, and writing—original draft, review, and editing.
SP Singh: conceptualization, supervision, funding acquisition, project administration, and writing—original draft, review, and editing.

## Conflict of Interest Statement

The authors declare that they have no conflict of interest.

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
