## [Reviewer comments · Life Science Alliance]

Life Science Alliance

Starvation resistant cavefish reveal conserved mechanisms of starvation-induced hepatic lipotoxicity

Macarena Pozo-Morales, Ansa Cobham, Cielo Centola, Mary McKinney, Peiduo Liu, Camille Perazzolo, Anne Lefort, Frédéric Libert, Hua Bai, Nicolas Rohner, and Sumeet Singh

DOI: <https://doi.org/10.26508/lsa.202302458>

Corresponding author(s): Sumeet Singh, Université Libre de Bruxelles and Nicolas Rohner, Stowers Institute for Medical Research

Review Timeline:

Submission Date:	2023-10-26
Editorial Decision:	2023-12-06
Revision Received:	2024-02-09
Editorial Decision:	2024-02-14
Revision Received:	2024-02-14
Accepted:	2024-02-19

Transaction Report:

December 6, 2023

Re: Life Science Alliance manuscript #LSA-2023-02458-T

Sumeet Pal Singh
Université Libre de Bruxelles
Belgium

Dear Dr. Singh,

Thank you for submitting your manuscript entitled "Starvation resistant cavefish reveal conserved mechanisms of starvation-induced hepatic lipotoxicity" to Life Science Alliance. The manuscript was assessed by expert reviewers, whose comments are appended to this letter. We invite you to submit a revised manuscript addressing the Reviewer comments.

Thank you for this interesting contribution to Life Science Alliance. We are looking forward to receiving your revised manuscript.

Sincerely,

B. MANUSCRIPT ORGANIZATION AND FORMATTING:

Reviewer #1 (Comments to the Authors (Required)):

This excellent paper by Pozo-Morales and colleagues employ an elegant comparative biology approach to study a highly relevant condition affecting health on a global scale: namely starvation-induced hepatic lipotoxicity. Based on the high degree of conservation of the phenomenon of starvation-induced steatosis, the authors compare cave and surface-dwelling populations of the Mexican tetra, *Astyanax mexicanus*, as well as the zebrafish and drosophila and use a combination of imaging and genetic tools, thereby implicating the fatty acid transporter *slc27a2a* in the regulation of fat processing in the liver during nutrition deprivation. The long-term practical benefit of this work is that it has identified a conserved druggable target for potential new therapies aimed at alleviating the pathological effects of starvation on liver function.

The data and images presented are all of very high quality and the analysis of the results as well as the conclusions which have been drawn are all totally reasonable. I have only minor suggestions to improve the presentation of the work for non-experts:

Line 69: Some more precise additional information would be beneficial to explain the availability of food in the cave environment. Referring to "extreme starvation" seems a little strange - since starvation itself is an extreme state (no food intake). It is not that there is absolutely no food available in the cave systems, more that while there is no plant life in the caves, limited food is washed into the caves from the outside world sporadically. This has led to various physiological adaptations aimed at exploiting a limited and non-regular food supply. All these points should be presented in more detail.

The discussion in the manuscript is extremely brief - (I am guessing constrained by the formatting rules of the journal). If at all possible, it would be beneficial to explore in more detail the implications of this work - for example in the more general context of pharmacological treatments aimed to influence fat metabolism by the liver... The implications of this work are far reaching - a point that doesn't come out in the current discussion.

Reviewer #2 (Comments to the Authors (Required)):

This is a confirmation study of ref. 14, and should undergo revisions aimed at placing the results in that context.

1. The authors replicate the results of ref. 14, but don't clearly state as such: in (now) two species of teleost fish, the FATP family member *SLC27a2a* is induced in livers during starvation. The differential expression among cave fish and surface fish is novel, as is the demonstration in *Drosophila*; however, the general replication nature of the study requires clear statement among the paragraphs covered by lines 194-208.
2. The phrase "liver injury" appears in the running title, but is used only once in the text (line 204). The authors are urged to remove this running title and the one appearance in the text. Starvation causes liver failure, a qualitatively different effect, and using 'failure' or 'death' removes the ambiguous association of the study with the hepatic steatosis with the progressive (recently renamed from "non-alcoholic") metabolic dysfunction associated liver disease. The macrophage infiltration since in figure 2 nicely reflects a profound evidence of dying (starving) hepatocytes being swallowed by these "big eater" immune cells, and the authors rightly avoid discussing MASLD.
3. The authors should assess whether treatment with the FATP2 inhibitor in Figure 3 prolongs life in starved larvae. The authors' seeming hypothesis is that as the animal undergoes whole-body autophagy and macromolecule (nutrient) recycling to liver, there is an excess accumulation of neutral lipids in liver that damages the liver (no histological or electron micrographical evidence of this, of course) and that also prevents feeding other organs with the trapped carbon atoms.
4. The authors should genotype their zebrafish to confirm the spontaneous *slc16a6a* mutation is absent. As reviewed in <https://link.springer.com/article/10.1007/s00018-012-1037-y>, this mutation dramatically increases fasting steatosis in zebrafish larvae and confounds much of zebrafish liver lipid metabolic research.
5. The authors should speculate on the molecular mechanisms accounting for differences between surface and cave fish.

Relating to reviewer 3's final point: the lipids from the yolk are exhausted by day 5-6 post-fertilization. The lipids that accumulate in the liver of never-fed animals can reflect re-esterified unspent (neither oxidized nor secreted as free fatty acids or as VLDL) fatty acyl chains from the yolk and from de novo lipogenesis (using carbon atoms from sugars and amino acids).

Reviewer #3 (Comments to the Authors (Required)):

In an effort to better understand the mechanisms underlying the liver lipotoxicity under starved conditions, Pozo-Morales et. al. use the starvation resistant *Astyanax mexicanus* cavefish and its starvation sensitive surface relatives, as well as zebrafish, to identify *slc27a2a* as critical for the lipotoxic response. Further, this gene also promotes lipotoxicity in flies, suggesting an evolutionarily conserved mechanism. The authors postulate that limiting fatty acid levels in the liver by disrupting *slc27a2/FATP2* mediated fatty acid transport protects the liver during starvation conditions.

The manuscript was clearly written and explores an important cellular process (lipotoxicity) by taking advantage of the unique cavefish model. The paper will likely be of interest to the readers of Life Science Alliance.

The following are a mix of major and minor concerns:

- 1) Fig1D: The authors characterize the liver starvation response in surface and cave species by measuring liver size and lipid droplet formation. Data on liver size was analyzed at 6, 8 and 12 dpf. However, lipid droplet data was only reported at 6 and 8 dpf. This raised the question: do cave fish develop lipid droplets in the liver at 12 dpf. If not, it raises the possibility that cave fish are not capable of forming liver lipid droplets. Are there any conditions (ie: longer period of starvation) known to the authors under which cavefish develop liver lipid droplets? Addressing this question will help to clarify whether there is a fundamental difference in liver starvation response or if it happens on a different time scale. This may be particularly important considering all fish assayed in this study are in larval stages and are undergoing development which could be a confounding factor.
- 2) Fig. 2A/B: please include liver images from a fed control for 6-8dpf to account for any developmental changes in macrophage infiltration and lipid droplet formation
- 3) Fig. 3F: zebrafish also express *slc27a2b* and a flanking tandem duplication *zgc:158482* on chromosome 18 , to what degree did the authors explore the regulation of these paralogs in the context of starvation? It is not uncommon that gene paralogs are coregulated, especially those derived from the whole genome wide duplication. These analyses should be included and discussed.
- 4) Fig. 3G: the authors use an *FATP2* inhibitor, *lipofermata*, to test the role of *slc27a2a* in promoting liver toxicity under starvation and show that inhibition reduces lipid droplet and liver atrophy. Increased inflammation and decreased life span are also components of the starvation response. Are these phenotypes also rescued with *lipofermata* treatment? These analyses should be included and discussed.
- 5) Fig. 3G-H: *lipofermata* presumably also impacts *slc27a2* paralog activity and may have other unknown off target effects. It is essential to include images of whole treated animals to exclude gross toxicity / altered development due to drug treatment. Data from fed animals +/- drug treatment should also be added - this is particularly important for interpreting the increased liver size observed in *lipofermata* treated animals. Images of whole larva will also reveal changes in yolk utilization which would impact the amount of lipid available for liver uptake.
- 6) These findings raise the question - during starvation, in the absence of functional *FATP2*, where does the lipid go if it no longer goes to the liver? The authors do not provide much of an accounting of the acyl chains. Since yolk lipid lipolysis is the source of much of these fatty acyl chains, it would be useful to see if the yolk mass is equivalent / absorbed equally in *lipofermata* treated animals?

Reviewer Cross Check - we would like to emphasize a few points made by the other reviewers:

The discussion is brief and that the manuscript would benefit from a more extended discussion of the implications of the work (Reviewer 1)

Replication of experiments from ref. 14 should be better stated in the text (Reviewer 2)

Zebrafish should be genotyped for the *slc16a16a* polymorphism (Reviewer 2)

Starvation resistant cavefish reveal conserved mechanisms of starvation-induced hepatic lipotoxicity.

Authors: Macarena Pozo-Morales^{1†}, Ansa E Cobham^{2†}, Cielo Centola², Mary Cathleen McKinney², Peiduo Liu³, Camille Perazzolo¹, Anne Lefort¹, Frédéric Libert¹, Hua Bai³, Nicolas Rohner^{2,4*}, Sumeet Pal Singh^{1*}

† These authors contributed equally.

* These authors co-supervised the work and contributed equally. Email for correspondence: nro@stowers.org and sumeet.pal.singh@ulb.be.

We thank the reviewers for the encouraging comments. We have attempted to address all the issues outlined by the reviewers. We thank the reviewers for raising the concerns, which were all valid. Addressing the comments has, we think, strengthened the manuscript, and for this we appreciate the constructive feedback received from the reviewers.

In the Response Letter, the following pattern is used:

Reviewer's comments are in black.

Our reply to reviewer's comments is in blue.

"Edits copied from the revised version of the manuscript are in quotes."

Reviewer #1 (Comments to the Authors (Required)):

This excellent paper by Pozo-Morales and colleagues employ an elegant comparative biology approach to study a highly relevant condition affecting health on a global scale: namely starvation-induced hepatic lipotoxicity. Based on the high degree of conservation of the phenomenon of starvation-induced steatosis, the authors compare cave and surface-dwelling populations of the Mexican tetra, *Astyanax mexicanus*, as well as the zebrafish and drosophila and use a combination of imaging and genetic tools, thereby implicating the fatty acid transporter *slc27a2a* in the regulation of fat processing in the liver during nutrition deprivation. The long-term practical benefit of this work is that it has identified a conserved druggable target for potential new therapies aimed at alleviating the pathological effects of starvation on liver function.

The data and images presented are all of very high quality and the analysis of the results as well as the conclusions which have been drawn are all totally reasonable. I have only minor suggestions to improve the presentation of the work for non-experts:

Line 69: Some more precise additional information would be beneficial to explain the availability of food in the cave environment. Referring to "extreme starvation" seems a little strange - since starvation itself is an extreme state (no food intake). It is not that there is absolutely no food available in the cave systems, more that while there is no plant life in the caves, limited food is washed into the caves from the outside world sporadically. This has led to various physiological adaptations aimed at exploiting a

limited and non-regular food supply. All these points should be presented in more detail.

We have provided a detailed explanation of the cavefish ecology and how that impacts the starvation response. We have removed the term “extreme starvation” and provided specific details related to the feeding regimen of the Mexican cavefish in the natural habitat, as follows:

“To gain mechanistic insights, we used a genetically tractable and naturally occurring model of starvation resistance - the *Astyanax mexicanus* model system (Cobham & Rohner, 2024; Jeffery, 2020). *A. mexicanus* are teleost freshwater fish found in the Sierra de El Abra region of Mexico. A unique feature of *A. mexicanus* is the presence of the surface and cavefish morphs that provides an ideal comparative model system for the study of starvation response. The surface populations reside in the rivers and have constant access to nutrition. In contrast, the cave populations of this species have adapted to survive in the nutrient poor subterranean environment, which provides limited nutrition in the dry season. Seasonal flooding during the rainy season brings nutrition inside the cave environment, leading to large fluctuations in food availability during the year (Wilson *et al*, 2021; Medley *et al*, 2022). Thus, while the surface morphs display relatively normal vertebrate physiology, the cave morphs have adapted to conditions of nutrition variation, including resistance to long periods of starvation.”

The discussion in the manuscript is extremely brief - (I am guessing constrained by the formatting rules of the journal). If at all possible, it would be beneficial to explore in more detail the implications of this work - for example in the more general context of pharmacological treatments aimed to influence fat metabolism by the liver... The

implications of this work are far reaching - a point that doesnt come out in the current discussion.

The discussion section has been expanded and we have included potential implications and the shortcomings of the study.

Reviewer #2 (Comments to the Authors (Required)):

This is a confirmation study of ref. 14, and should undergo revisions aimed at placing the results in that context.

1. The authors replicate the results of ref. 14, but don't clearly state as such: in (now) two species of telost fish, the FATP family member SLC27a2a is induced in livers during starvation. The differential expression among cave fish and surface fish is novel, as is the demonstration in *Drosophila*; however, the general replication nature of the study requires clear statement among the paragraphs covered by lines 194-208.

We have noted the findings from *Xu et al.* (referred to as ref. 14) in the Discussion section. However, please note that in parallel to *Xu et al.*, we have also published on the starvation-induced hepatic steatosis in zebrafish larvae, and linked it with cytoplasmic calcium flux (*Pozo-Morales et al., 2023*). In the current manuscript, we extend this topic and identify cavefish as a unique animal model that does not display the evolutionary-conserved phenomenon of starvation-induced hepatic steatosis.

Further, as noted in the revised Discussion section, *Xu et al.* postulated CD36 as a regulator of starvation-induced hepatic steatosis. However, the expression of CD36 is an order of magnitude lower than the expression of *slc27a2a* in the zebrafish liver (Fig 3F). Comparative data from the *Astyanax mexicanus* model system shows the same trend in the expression levels of CD36 and *slc27a2a* (FATP2) (Fig 3F). Further, there is no direct homologue for CD36 in *Drosophila*. Genes belonging to the CD36 family of lipid receptors, called *Croquemort* (*Crq*)

(Guillou *et al*, 2016) and Sensory neuron membrane protein (SNMP) (Benton *et al*, 2007) are found in *Drosophila*. However, their capacity to induce fatty acid uptake is not known. In contrast, our results indicate that FATP2 is essential for starvation-induced steatosis in fly oenocytes.

Thus, we propose FATP2, and not CD36, to be the evolutionarily conserved regulator of starvation-induced hepatic steatosis.

2. The phrase "liver injury" appears in the running title, but is used only once in the text (line 204). The authors are urged to remove this running title and the one appearance in the text. Starvation causes liver failure, a qualitatively different effect, and using 'failure' or 'death' removes the ambiguous association of the study with the hepatic steatosis with the progressive (recently renamed from "non-alcoholic") metabolic dysfunction associated liver disease. The macrophage infiltration since in figure 2 nicely reflects a profound evidence of dying (starving) hepatocytes being swallowed by these "big eater" immune cells, and the authors rightly avoid discussing MASLD.

This is an important point raised by the reviewer. We have changed "liver injury" to 'liver atrophy'. As rightly pointed by the reviewer, our study looks at the size of the liver and the clearance of dead hepatocytes as a proxy for atrophy.

3. The authors should assess whether treatment with the FATP2 inhibitor in Figure 3 prolongs life in starved larvae. The authors' seeming hypothesis is that as the animal undergoes whole-body autophagy and macromolecule (nutrient) recycling to liver, there is an excess accumulation of neutral lipids in liver that damages the liver (no

histological or electron micrographical evidence of this, of course) and that also prevents feeding other organs with the trapped carbon atoms.

We generated the survival curve of animals treated with lipofermata and did not see differences in starvation resistance (Fig S5E). This suggests additional mechanisms of increased starvation resistance might play a role in cavefish. Of note, as pointed in response to point 5 below, cavefish larvae have been demonstrated to display enhanced metabolic plasticity that enables them to enter a low-energy state upon starvation (Bilandžija *et al*, 2020). This could be an important factor in improving starvation resistance.

Nevertheless, our study highlights that cavefish larvae are an ideal model system to study aspects of starvation resistance and associated organ atrophy.

4. The authors should genotype their zebrafish to confirm the spontaneous *slc16a6a* mutation is absent. As reviewed in <https://link.springer.com/article/10.1007/s00018-012-1037-y>, this mutation dramatically increases fasting steatosis in zebrafish larvae and confounds much of zebrafish liver lipid metabolic research.

We have genotyped our experimental zebrafish and find that they do not harbor the mutation in *slc16a16a* (Fig S1).

5. The authors should speculate on the molecular mechanisms accounting for differences between surface and cave fish.

The reviewer raises two critical points: What regulates *slc27a2a* expression levels upon starvation? And why is this regulation lost in cavefish?

Currently, we do not know the upstream regulators of *slc27a2a*, and very limited information exists in the literature on this topic. The regulation could be endocrine, controlled by the increased glucagon secretion from pancreatic alpha-cells during fasting. Ongoing studies in the lab indicate that genetic ablation of pancreatic alpha-cell identity reduces starvation resistance in zebrafish. However, as the work is highly preliminary, and we do not know if the manipulation regulates *slc27a2a* expression levels, we cannot assign specific upstream regulators of *slc27a2a*. However, the ability to inbreed surface and cavefish allows, in principle, to perform quantitative trait locus (QTL) analysis to map genetic loci regulating the phenotype.

Thus, we have added potential speculative mechanisms and the experimental possibilities to identify them using QTL mapping in the Discussion Section as follows:

“Our study, however, does not identify the molecular link between starvation and enhancement of *slc27a2a* expression. Endocrine signaling could be a potential regulator of initiating starvation response in the liver. Hepatocytes are sensitive to insulin and glucagon, which are released from the pancreas in response to feeding and fasting respectively. Changes in the levels of circulating insulin and / or glucagon hormones could be sensed as a fed-to-fast transition in the hepatocytes.

Accordingly, work in *Drosophila* has identified a crosstalk between the fat body and oenocytes via the insulin signaling pathway as being responsible for starvation-induced accumulation of lipid droplets (Chatterjee *et al*, 2014). Notably, cavefish carry a hypomorphic mutation in insulin receptor (Riddle *et al*, 2018), which could reduce its ability to sense such a fed-to-fast transition. In future, it would be of interest to investigate the link between insulin signaling and *slc27a2a* expression

levels. Further, the search for regulators of starvation-induced hepatic steatosis can be aided by performing Quantitative Trait Loci (QTL) mapping in *A. mexicanus* (Stockdale *et al*, 2018; O'Quin *et al*, 2013). This is made possible by the fact that cavefish and surface fish are inter-fertile, which allows generation of cave / surface genetic hybrids, providing a unique platform to identify potential regulators of the phenomenon.”

Relating to reviewer 3's final point: the lipids from the yolk are exhausted by day 5-6 post-fertilization. The lipids that accumulate in the liver of never-fed animals can reflect re-esterified unspent (neither oxidized nor secreted as free fatty acids or as VLDL) fatty acyl chains from the yolk and from de novo lipogenesis (using carbon atoms from sugars and amino acids).

As noted in the response to Reviewer #3, it could be postulated that the lack of absorption of the lipids into the liver might lead to an increase of triglycerides and free fatty acids in the blood (hyperlipidemia). However, owing to its small size, our model of fish larvae lacks the possibility to measure lipid levels in the blood. An alternative is to utilize reporters of lipids in the circulation, for instance the LipoGlo system from the Farber Lab (Thierer *et al*, 2019). However, we currently do not have access of this fish, and were unable to test its response to lipofermata treatment.

We are currently generating the *slc27a2a* (*fatp2*) zebrafish mutant. In this animal, it would be of interest to measure blood lipid levels under homeostasis and starvation condition in the adults, where a higher quantity of blood can be collected. However, adult zebrafish can survive without food for more than a month (personal observation); thus, we still need to setup the starvation protocol in adults.

Reviewer #3 (Comments to the Authors (Required)):

In an effort to better understand the mechanisms underlying the liver lipotoxicity under starved conditions, Pozo-Morales et. al. use the starvation resistant *Astyanax mexicanus* cavefish and its starvation sensitive surface relatives, as well as zebrafish, to identify *slc27a2a* as critical for the lipotoxic response. Further, this gene also promotes lipotoxicity in flies, suggesting an evolutionarily conserved mechanism. The authors postulate that limiting fatty acid levels in the liver by disrupting *slc27a2/FATP2* mediated fatty acid transport protects the liver during starvation conditions.

The manuscript was clearly written and explores an important cellular process (lipotoxicity) by taking advantage of the unique cavefish model. The paper will likely be of interest to the readers of Life Science Alliance.

The following are a mix of major and minor concerns:

1) Fig1D: The authors characterize the liver starvation response in surface and cave species by measuring liver size and lipid droplet formation. Data on liver size was analyzed at 6, 8 and 12 dpf. However, lipid droplet data was only reported at 6 and 8 dpf. This raised the question: do cave fish develop lipid droplets in the liver at 12 dpf. If not, it raises the possibility that cave fish are not capable of forming liver lipid droplets. Are there any conditions (ie: longer period of starvation) known to the authors under which cavefish develop liver lipid droplets? Addressing this question will help to clarify whether there is a fundamental difference in liver starvation response or if it happens on a different time scale. This may be particularly important

considering all fish assayed in this study are in larval stages and are undergoing development which could be a confounding factor.

We have included the images (Fig 1D) and quantification (Fig 1B) of lipid droplets in the livers of cavefish and surface fish. We find no increase in lipid droplets in the cavefish at 12 dpf, suggesting that the animals do not accumulate lipid droplets in the liver even upon extending the fasting period.

We have not conducted experiments to test if cavefish accumulate lipid droplets in response to a high-fat / high-cholesterol diet. However, this could certainly be of interest to test in future.

2) Fig. 2A/B: please include liver images from a fed control for 6-8dpf to account for any developmental changes in macrophage infiltration and lipid droplet formation
In our previous study on the role of calcium in starvation-induced hepatic steatosis, we demonstrated that feeding resolves lipid droplets in the liver (Fig R1) (Pozo-Morales *et al*, 2023).

Figure R1: Resolution of hepatic steatosis upon feeding. (A) Maximum Intensity Projection (MIP) of livers from 7 dpf fasted and fed animals. (B) Barplot comparing the normalized lipid droplets in the hepatocytes of 7 dpf fasted and fed

animals. *** p-value < 0.001, Student's t-test.

Figure adapted from (Pozo- Morales *et al*, 2023).

In the revised version, we have included the image and analysis of macrophage infiltration in the liver of fed animals and find that it is at baseline levels (Fig S2). This suggests that macrophage infiltration is a characteristic of the starvation response.

3) Fig. 3F: zebrafish also express *slc27a2b* and a flanking tandem duplication *zgc:158482* on chromosome 18 , to what degree did the authors explore the regulation of these paralogs in the context of starvation? It is not uncommon that gene paralogs are coregulated, especially those derived from the whole genome wide duplication. These analyses should be included and discussed.

Both *slc27a2b* and *zgc:158482* do not display significant changes in gene expression upon starvation. Further, they are expressed at a very low level (< 10 Reads Per-Million) in the liver.

4) Fig. 3G: the authors use an FATP2 inhibitor, lipofermata, to test the role of *slc27a2a* in promoting liver toxicity under starvation and show that inhibition reduces lipid droplet and liver atrophy. Increased inflammation and decreased life span are also components of the starvation response. Are these phenotypes also rescued with lipofermata treatment? These analyses should be included and discussed.

We agree with the reviewer that these phenotypes should be described in detail. To this end, we performed bulk RNA-Sequencing on livers isolated from animals treated

with vehicle or lipofermata. This allowed us to obtain an unbiased insight into the impact of lipofermata (*slc27a2* inhibitor) on the liver.

We describe the experiment in the Result section:

“To identify the genes regulated by the lipofermata treatment, we performed differential gene expression analysis of livers collected from fasting 6 dpf zebrafish treated with vehicle or lipofermata (Fig S4A). The analysis identified an up-regulation of 843 genes and a down-regulation of 1067 genes in the liver upon lipofermata treatment (Table S1). Notably, genes related to cell-cycle, including cyclins (*ccna2*, *ccnb1*, *ccnb2*, *ccnb3*), Ki-67 (*mki67*) and polo-like kinase 1 (*plk1*) were up-regulated upon lipofermata treatment (Fig S4B). The increase in cell-cycle related genes corroborates the increase in liver size in lipofermata treated animals. Further, lipofermata treatment led to a reduction in the expression of cytokines, including TNF superfamily member 10 (*tnfsf10*), chemokine ligands (*ccl25b*, *cxcl12a*, *cxcl18b*) and BMP / TGF-beta pathway ligands (*bmp2a*, *tgfb1a*, *tgfb1b*) (Fig S4C), suggesting a lowering of the inflammatory molecules in the milieu. However, the number of macrophages in the liver did not change upon lipofermata treatment (Fig S4D). Overall, the transcriptional profiling of livers from lipofermata treated animals suggests that the organ can initiate cell replication in the fasting state, thereby being capable of maintaining its size.”

We further extend the implication of this result in the Discussion section:

“Notably, the antagonism of *slc27a2a* increases expression of cell-cycle related genes in the liver, while reducing the expression of inflammatory cytokines (Fig S4B, C). Particularly, the treatment reduces the expression of TGF- β ligands, which have been shown to induce hepatic fibrosis, a key step in irreversible liver damage

(Dooley & ten Dijke, 2012). Thus, our study shows that the optimal solution to starvation-induced liver atrophy, identified by naturally starvation-resistant cavefish, is to prevent the accumulation of hepatic lipid droplets.”

Further, we generated the survival curve of animals treated with lipofermata and did not see differences in starvation resistance (Fig S5E). This suggests additional mechanisms of increased starvation resistance might play a role in cavefish, which we discuss as follows:

“Though inhibition of *slc27a2a* can protect the zebrafish liver from atrophy, it does not improve the organism’s starvation resistance (Fig S5E). Metabolic plasticity in other organs might play a critical role in the enhanced starvation resistance of cavefish larvae (Fig 1A). It is known that cavefish larvae reduce their average oxygen consumption upon starvation (Bilandžija *et al*, 2020). In *Drosophila*, enhanced starvation resistance is related to changes in basal metabolic rate (Brown *et al*, 2019). Future investigations in the cavefish’s ability to survive long period of famine are required; however, our work does highlight the utility of comparative evolutionary analysis in uncovering regulators of organ atrophy upon starvation.”

5) Fig. 3G-H: lipofermata presumably also impacts *slc27a2* paralog activity and may have other unknown off target effects. It is essential to include images of whole treated animals to exclude gross toxicity / altered development due to drug treatment. Data from fed animals +/- drug treatment should also be added - this is particularly important for interpreting the increased liver size observed in lipofermata treated animals. Images of whole larva will also reveal changes in yolk utilization which would impact the amount of lipid available for liver uptake.

This is a critical point, and we conducted the experiments as proposed by the reviewer. We observe no morphological defects in the treated animals (Fig S5A) or impact of the treatment on body length (Fig S5B). Further, treatment of fed animals with lipofermata did not show any significant increase in liver size (Fig S5C, D).

6) These findings raise the question - during starvation, in the absence of functional FATP2, where does the lipid go if it no longer goes to the liver? The authors do not provide much of an accounting of the acyl chains. Since yolk lipid lipolysis is the source of much of these fatty acyl chains, it would be useful to see if the yolk mass is equivalent / absorbed equally in lipofermata treated animals?

We have imaged the zebrafish larvae treated with lipofermata and do not find persistence of yolk in the animals (Fig S5A). It could be postulated that the lack of absorption of the lipids into the liver might lead to an increase of lipids in the blood (hyperlipidemia). However, owing to its small size, our model of fish larvae lacks the possibility to measure lipid levels in the blood. An alternative is to utilize reporters of lipids in the circulation, for instance the LipoGlo system from the Farber Lab (Thierer *et al*, 2019). However, we currently do not have access of this fish, and were unable to test its response to lipofermata treatment.

We are currently generating the *slc27a2a* (*fatp2*) zebrafish mutant. In this animal, it would be of interest to measure blood lipids levels under homeostasis and starvation condition in the adults, where a higher quantity of blood can be collected. However, adult zebrafish can survive without food for more than a month (personal observation); thus, we still need to setup the starvation protocol in adults.

Reviewer Cross Check - we would like to emphasize a few points made by the other

reviewers:

The discussion is brief and that the manuscript would benefit from a more extended discussion of the implications of the work (Reviewer 1)

Detailed Discussion section has been added to the revised manuscript.

Replication of experiments from ref. 14 should be better stated in the text (Reviewer 2)

We have noted the findings from *Xu et al.* (referred to as ref. 14) in the Discussion section. However, please note that in parallel to *Xu et al.*, we have also published on the starvation-induced hepatic steatosis in zebrafish larvae, and linked it with cytoplasmic calcium flux (Pozo- Morales *et al*, 2023). In the current manuscript, we extend this topic and identify cavefish as a unique animal model that does not display the evolutionary-conserved phenomenon of starvation-induced hepatic steatosis.

Zebrafish should be genotyped for the *slc16a16a* polymorphism (Reviewer 2)

We have genotyped our experimental zebrafish and find that they do not harbor the mutation in *slc16a16a* (Fig S1).

References:

Benton R, Vannice KS & Vosshall LB (2007) An essential role for a CD36-related receptor in pheromone detection in *Drosophila*. *Nature* 450: 289–293

Bilandžija H, Hollifield B, Steck M, Meng G, Ng M, Koch AD, Gračan R, Četković H, Porter ML, Renner KJ, *et al* (2020) Phenotypic plasticity as a mechanism of cave colonization and adaptation. *eLife* 9: e51830

- Brown EB, Slocumb ME, Szuperak M, Kerbs A, Gibbs AG, Kayser MS & Keene AC (2019) Starvation resistance is associated with developmentally specified changes in sleep, feeding and metabolic rate. *J Exp Biol* 222: jeb191049
- Chatterjee D, Katewa SD, Qi Y, Jackson SA, Kapahi P & Jasper H (2014) Control of metabolic adaptation to fasting by dILP6-induced insulin signaling in *Drosophila oenocytes*. *Proceedings of the National Academy of Sciences* 111: 17959–17964
- Cobham AE & Rohner N (2024) Unraveling stress resilience: Insights from adaptations to extreme environments by *Astyanax mexicanus* cavefish. *J Exp Zool B Mol Dev Evol*
- Dooley S & ten Dijke P (2012) TGF- β in progression of liver disease. *Cell Tissue Res* 347: 245–256
- Guillou A, Troha K, Wang H, Franc NC & Buchon N (2016) The *Drosophila* CD36 Homologue croquemort Is Required to Maintain Immune and Gut Homeostasis during Development and Aging. *PLoS Pathog* 12: e1005961
- Jeffery WR (2020) *Astyanax* surface and cave fish morphs. *EvoDevo* 11: 14
- Medley JK, Persons J, Biswas T, Olsen L, Peuß R, Krishnan J, Xiong S & Rohner N (2022) The metabolome of Mexican cavefish shows a convergent signature highlighting sugar, antioxidant, and Ageing-Related metabolites. *eLife* 11: e74539
- O'Quin KE, Yoshizawa M, Doshi P & Jeffery WR (2013) Quantitative genetic analysis of retinal degeneration in the blind cavefish *Astyanax mexicanus*. *PLoS One* 8: e57281
- Pozo- Morales M, Garteizgogeoasca I, Perazzolo C, So J, Shin D & Singh SP (2023) In vivo imaging of calcium dynamics in zebrafish hepatocytes. *Hepatology* 77: 789
- Riddle MR, Aspiras AC, Gaudenz K, Peuß R, Sung JY, Martineau B, Peavey M, Box AC, Tabin JA, McGaugh S, *et al* (2018) Insulin resistance in cavefish as an adaptation to a nutrient-limited environment. *Nature* 555: 647–651
- Stockdale WT, Lemieux ME, Killen AC, Zhao J, Hu Z, Riepsaame J, Hamilton N, Kudoh T, Riley PR, van Aerle R, *et al* (2018) Heart Regeneration in the Mexican Cavefish. *Cell Rep* 25: 1997-2007.e7
- Thierer JH, Ekker SC & Farber SA (2019) The LipoGlo reporter system for sensitive and specific monitoring of atherogenic lipoproteins. *Nat Commun* 10: 3426
- Wilson EJ, Tobler M, Riesch R, Martínez-García L & García-De León FJ (2021) Natural history and trophic ecology of three populations of the Mexican cavefish, *Astyanax mexicanus*. *Environmental Biology of Fishes* 104: 1461–1474

February 14, 2024

RE: Life Science Alliance Manuscript #LSA-2023-02458-TR

Dr. Sumeet Pal Singh
IRIBHM, ULB
808 route de Lennik
Campus Erasme, Building C
Brussels 1070
Belgium

Dear Dr. Singh,

Thank you for submitting your revised manuscript entitled "Starvation resistant cavefish reveal conserved mechanisms of starvation-induced hepatic lipotoxicity". We would be happy to publish your paper in Life Science Alliance pending final revisions necessary to meet our formatting guidelines.

- please be sure that the authorship listing and order is correct
- please add ORCID ID for the secondary corresponding author -- they should have received instructions on how to do so
- we encourage you to revise the figure legend for figure S4 such that the figure panels are introduced in an alphabetical order
- please add a callout for Figure S2A to your main manuscript text
- the contributions selected for author Hua Bai do not qualify them for authorship. Please either update the contributions in our system and in the Author Contributions section of the manuscript, or let us know if this author should be removed.

A. FINAL FILES:

B. MANUSCRIPT ORGANIZATION AND FORMATTING:

Sincerely,

Reviewer #2 (Comments to the Authors (Required)):

The authors have addressed my concerns, strengthening their conclusions and setting the stage for their future work.

Reviewer #3 (Comments to the Authors (Required)):

The authors have satisfactorily addressed our concerns

February 19, 2024

RE: Life Science Alliance Manuscript #LSA-2023-02458-TRR

Dr. Sumeet Pal Singh
Université Libre de Bruxelles
808 route de Lennik
Campus Erasme, Building C
Brussels 1070
Belgium

Dear Dr. Singh,

Thank you for submitting your Research Article entitled "Starvation resistant cavefish reveal conserved mechanisms of starvation-induced hepatic lipotoxicity". It is a pleasure to let you know that your manuscript is now accepted for publication in Life Science Alliance. Congratulations on this interesting work.

DISTRIBUTION OF MATERIALS:

Again, congratulations on a very nice paper. I hope you found the review process to be constructive and are pleased with how the manuscript was handled editorially. We look forward to future exciting submissions from your lab.

Sincerely,
